# Experimental Research and Numerical Modelling of the Cold Forming Process of the Inconel 625 Alloy Sheets Using Flexible Punch

**DOI:** 10.3390/ma17010085

**Published:** 2023-12-23

**Authors:** Maciej Balcerzak, Krzysztof Żaba, Marcin Hojny, Sandra Puchlerska, Łukasz Kuczek, Tomasz Trzepieciński, Vit Novák

**Affiliations:** 1Department of Metal Working and Physical Metallurgy of Non-Ferrous Metals, Faculty of Non-Ferrous Metals, AGH University of Krakow, al. Adama Mickiewicza 30, 30-059 Cracow, Polandspuchler@agh.edu.pl (S.P.); lukasz.kuczek@agh.edu.pl (Ł.K.); 2Department of Applied Computer Science and Modelling, Faculty of Metal Engineering and Industrial Computer Science, AGH University of Krakow, al. Adama Mickiewicza 30, 30-059 Cracow, Poland; mhojny@agh.edu.pl; 3Department of Manufacturing Processes and Production Engineering, Faculty of Mechanical Engineering and Aeronautics, Rzeszow University of Technology, al. Powst. Warszawy 8, 35-959 Rzeszów, Poland; tomtrz@prz.edu.pl; 4Department of Manufacturing Technology, Faculty of Mechanical Engineering, Czech Technical University in Prague, Technická 4, 166 07 Prague 6, Czech Republic; vit.novak@fs.cvut.cz

**Keywords:** elastomeric materials, finite element method, Inconel 625, nickel alloys, sheet metal forming

## Abstract

The paper presents the numerical and experimental results of research aimed at determining the influence of hardness in the range of 50–90 Shore A of layered tools composed of elastomeric materials on the possibility of forming Inconel 625 nickel-based alloy sheets. A stamping die composed of 90MnCrV8 steel (hardness 60HRC) was designed for forming embosses in drawpieces, ensuring various stress states on the cross-section of the formed element. The principle of operating the stamping die was based on the Guerin method. The finite-element-based numerical modelling of the forming process for various configurations of polyurethane inserts was also carried out. The drawpieces obtained through sheet forming were subjected to geometry tests using optical 3D scanning. The results confirmed that, in the case of forming difficult-to-deform Inconel 625 Ni-based alloy sheets, the hardness of the polyurethane inserts significantly affected the geometric quality of the obtained drawpieces. The assumptions determined in numerical simulations were verified in experimental studies. Based on the test results, it was concluded that the selection of polyurethane hardness should be determined by the shape of the formed element. Significant nonuniform sheet metal deformations were also found, which may pose a problem in the process of designing forming tools and the technology of the plastic forming of Inconel 625 Ni-based alloy sheets.

## 1. Introduction

For many years, sheets composed of difficult-to-deform alloys have been widely used in the aviation industry [1,2]. In addition to conventional methods, electromagnetic forming methods [3] and those using flexible tools [4,5] are used. This is dictated by the need to produce many elements with different geometries in strictly limited quantities. Aircraft design involves the use of many components that must meet detailed standard requirements regarding dimensional accuracy and strength [6]. The most common types of sheet metal that are formed through processes using elastomeric materials are aluminium [7,8] and aluminium alloys [9]. However, difficult-to-deform materials, such as high-strength steels [10], titanium alloys [11] or Ni-based superalloys [12,13], can be successfully formed. The use of this type of alloy to form processes with elastomeric materials causes many problems. The main limitation is the alloy’s inability to form at elevated temperatures, owing to the limits of the elastomer’s temperature resistance [14,15]. Moreover, obtaining drawpiece profiles with small edge-rounding radii is also difficult, owing to the springback phenomenon [16,17] and the limited formability of sheet metal [18].

The dies for sheet metal forming (SMF) using flexible punches can be composed of various materials, such as steel, aluminium, wood or plastics. The die material is selected based on the designer’s experience, the properties it is required to meet and economic factors. Dies composed of aluminium are most often used in industry as tools for processes using elastomeric materials for stamping [19]. This is due to a very good strength-to-weight ratio, which is very important because, in most cases, the dies are placed manually in the press working area. However, conventional steel dies are characterised by a high wear resistance and mechanical strength. They are used to produce elements that are produced in mass production due to the high cost of making the tool [20].

Elastomeric materials belong to the group of cross-linked amorphous polymers, which can deform by up to 600%. They also show good shape memory properties [21,22]. The basic materials included in the group of elastomers include natural rubber, synthetic rubbers, silicones and polyurethanes [23,24,25]. The flexible material chosen most often for SMF tools is polyurethane. It is highly resistant to wear and is thermally stable. Furthermore, it is resistant to contact with chemicals. An important aspect is that polyurethane has viscoelastic properties [26]. This causes it to behave like an incompressible liquid during SMF, and when parts are formed in a closed container, the elastomeric material exerts uniform pressure on the sheet metal. Although elastomeric materials are widely used in the SMF process, their usefulness in this process remains insufficiently researched [27,28,29]. In many studies [24,30,31], the aim of which was to analyse SMF processes using flexible punches, a significant advantage of polyurethane over other materials was proven.

The literature also contains publications related directly to the simulation of plastic-forming processes, in which elastomeric materials are used [32,33,34,35,36,37]. The authors of these publications focused on the numerical modelling of rubber forming [33,36], the flanging process [37] and stamping using a flexible punch [32,34,35]. The results presented in these publications confirmed the possibility of obtaining results very close to those determined experimentally. Some authors, however, focused on determining the properties of the process or drawpiece. Ramezani et al. [23] presented results on the springback of sheet metal during forming with the use of flexible tools, while Ali et al. [38] referred to the friction conditions occurring during the forming process. Elastomeric tools have found wide application in the rubber-pad process for forming microchannels [39,40], embossing [41,42] and the patterning of thin metallic plates [43].

In this work, the selection of Inconel-type nickel alloy sheets as materials intended for plastic forming tests using a polyurethane punch was dictated by their high strength and slowness in cold forming with metallic tools, while Inconel alloys are used widely in the aviation industry to produce components mounted in aircraft engines. Giving the appropriate shape to products composed of high-strength sheet metal means that the elastomeric material is operated on under high-contact pressure, which is necessary to form components from these materials. Previous reports, in the form of both scientific publications and the written experiences of aviation company employees, concluded that there are no clearly defined guidelines regarding the type and properties of elastomeric materials used for SMF for difficult-to-form materials. This gap became the motivation for conducting the research presented in this article. The aim of the research was to determine the influence of the hardness and properties of a polyurethane punch on the formability of Inconel 625 alloy sheets. Polyurethane inserts with a Shore A (ShA) hardness of 50, 60, 70, 80 and 90 were used for the forming. A specially fabricated Cr123M steel die allowed for the possibility to assess the impact of the state of stress in the formed component on the quality of the drawpieces produced using the Guerin process [28].

## 2. Materials and Methods

### 2.1. Test Materials

#### 2.1.1. Sheet Metal

To test the forming of drawpieces using a polyurethane punch, 1 mm thick sheets composed of an Inconel 62 Ni-based alloy were used. The basic mechanical properties were determined in a uniaxial tensile test in accordance with standard EN ISO 6892-1:2020-05 [44]. The shape and dimensions of the samples for testing the mechanical properties are shown in Figure 1.

A static uniaxial tensile test was performed using a Z100 (Zwick/Roell, Ulm, Germany) testing machine with a strain rate of 0.008 s^−1^. The basic mechanical parameters of the sheets were determined, including the ultimate tensile strength R_m_, yield stress R_p0.2_ and total elongation A. The analysis of the anisotropy of the sheets was also performed based on the uniaxial tensile test. Samples for testing were cut at angles of 0°, 90° and 45° relative to the sheet-rolling direction. The base test length was set to 25% of the sample elongation, which was determined from the tensile test up to the failure of the sample.

The surface roughness parameters and surface topography of the Inconel 625 sheet metal and polyurethane inserts were analysed using the LEXT OLS4100 (Olympus, Tokio, Japan) laser scanning digital microscope for noncontact three-dimensional (3D) observations and measurements of surface features at 10 nm resolutions. The microscope was equipped with a 405 nm short-wavelength semiconductor laser. The average values of the surface roughness parameters (average surface roughness Ra and ten-point height of irregularities Rz) were determined based on the three measurements of each sample. The average roughness Ra and ten-point height of irregularities Rz of the Inconel 625 sheet were equal to 0.105 μm and 0.646 μm, respectively. Figure 2 shows the surface topography and linear roughness profiles of the Inconel 625 nickel alloy samples.

#### 2.1.2. Material of Punch

Commercial polyurethanes with a nominal hardness of 50, 60, 70, 80 and 90 ShA were used for the tests. The test samples and inserts for the forming tool were created from one batch of material so that there were no significant differences in their properties. Hardness tests and uniaxial compression tests were carried out. The compression test of the polyurethane samples was carried out in accordance with the [45]. Cylindrical samples with a diameter of D = 28.6 mm and a height of h = 13 mm were prepared. The samples were compressed at a speed of 12 mm/min to a height of 10 mm. Compression tests were carried out without the use of lubricant. The research was carried out taking into account three repetitions of each test. Table 1 shows the results after measuring the basic roughness parameters of the polyurethane inserts.

### 2.2. Wear Resistance of Elastomeric Materials

The abrasive wear resistance of the polyurethane countersamples was tested on the T-05 roller-block tester (Institute of Precision Mechanics, Radom, Poland). The measurement was carried out at an ambient temperature in a dry sliding contact. The operating principle of the tester is shown schematically in Figure 3a.

The self-adjusting mounting of the block, which consisted of a sample holder and a hemispherical insert, ensured the good adhesion of the block to the roller and an even distribution of pressure on the contact surface. The tester enabled tests to be carried out in accordance with the methods specified in [46,47,48,49]. The geometry and dimensions of the polyurethane countersample are shown in Figure 3b. The sample was created with an Inconel 625 alloy.

All the measurements were performed at a constant ring rotation speed of *n* = 136 rpm. During the test, a load of *F*_N_ = 50 N was applied. The friction path was 150 m. During the tribological test, the friction force *F* was continuously recorded, which was used to determine the coefficient of friction (COF) *µ* according to Equation (1).
(1)$$\mu =\frac{F}{F_{N}}$$

The COF was determined as the average value for the entire friction path. The measure of abrasion resistance is the mass loss of the tested material in relation to the friction path and the applied load. The percentage weight loss Δ*m_cs_* of the polyurethane countersamples was determined according to Equation (2):(2)$${∆m}_{cs}=\frac{m_{p}-m_{k}}{m_{p}}\times 100\%$$
where *m_p_* is the initial mass of the countersample and *m_k_* is final mass of the countersample.

In a similar way, the percentage weight loss Δ*m_s_* of the sample was determined for a sample composed of Inconel 625.

### 2.3. Methodology of Numerical Simulations

Finite-element-based numerical simulations of the forming process of Inconel 625 sheets were carried out using Impetus Afea (Impetusafea AB, version 8.1, Huddinge, Sweden) software. The determination of the material coefficients of the polyurethane samples used in the simulations was based on a uniaxial compression test of a cylindrical sample, as mentioned in Section 2.1.2. The mechanical parameters of the Inconel 625 alloy sheets were determined on the basis of a uniaxial tensile test using a Z100 (Zwick/Roell, Ulm, Germany) uniaxial tensile testing machine. In the case of the elastomeric materials, a two-parameter Mooney–Rivlin constitutive model was selected:(3)$$\sigma_{1}=\frac{2C_{1}}{3}\left(2\lambda_{1}-\lambda_{2}-\lambda_{3}\right)-\frac{2C_{2}}{3}\left(\frac{2}{\lambda_{1}}-\frac{1}{\lambda_{2}}-\frac{1}{\lambda_{3}}\right)-\rho$$
(4)$$\sigma_{2}=\frac{2C_{1}}{3}\left(2\lambda_{2}-\lambda_{3}-\lambda_{1}\right)-\frac{2C_{2}}{3}\left(\frac{2}{\lambda_{2}}-\frac{1}{\lambda_{3}}-\frac{1}{\lambda_{1}}\right)-\rho$$
(5)$$\sigma_{3}=\frac{2C_{1}}{3}\left(2\lambda_{3}-\lambda_{1}-\lambda_{2}\right)-\frac{2C_{2}}{3}\left(\frac{2}{\lambda_{3}}-\frac{1}{\lambda_{1}}-\frac{1}{\lambda_{2}}\right)-\rho$$
(6)$$\rho =-K\varepsilon_{v}$$
where $\sigma_{1},\sigma_{2},\sigma_{3}$ are the principal stresses, $\lambda_{1},\lambda_{2},\lambda_{3}$ are the eigenvalues of Cauchy–Green’s right stretch tensor, the pressure ρ is a linear function of the volumetric strain and $\varepsilon_{v},$ ***C*_1_**, ***C*_2_** and ***K*** are the material constants.

The Mooney–Rivlin material model was used due to its availability in the software used to perform the numerical simulations. For each material, a 3D model was developed reflecting the sample geometry and test conditions. Simulations were carried out to determine the material coefficients *C*_1_ and *C*_2_ in such a way that the experimental results were reflected in the simulations as much as possible. An example result of the simulation of the sample compression process and the fitting of the simulation results to the experimental curve is presented in Figure 4a and Figure 4b, respectively.

Table 2 shows the material coefficients obtained from the simulation of the compression tests.

The coefficient of friction (μ = 0.09) was assumed to be constant in the simulation model. The same value of the coefficient of friction was used for the polyurethane inserts of different hardnesses due to the possibility of a direct comparison of the results. The constitutive model was designed as a temperature- and strain-rate-dependant strength model in a simple isotopic form. The von Mises yield criterion, the elastic–plastic law of plastic flow and explicit time integration scheme were used in the calculations. A contact algorithm based on a penalty function was used. Shell elements were used to in the discretization of the forming tools (die, stamp and container). The forming tools were considered to be rigid. The workpiece and polyurethane inserts were assumed as being deformable. An optimum number of elements and element size were determined based on the mesh sensitivity analysis (Figure 5).

After determining all the necessary material parameters, a simplified 3D model of the stamping die was created to simulate the sheet-metal-forming process using polyurethane punches (Figure 6). The simplification of the stamping die model due to the symmetry of the forming process aimed at minimising the computational time of the numerical simulations.

Several simulations were performed assuming different combinations of hardnesses of polyurethane inserts. Two variants were considered. In the first variant, a stack of five inserts with the same hardnesses were used. The simulations were also carried out assuming the use of hybrid variants of the arrangement of polyurethane inserts, where the use of a stack of polyurethane inserts of different hardnesses in one tool was considered 2 × 50 ShA + 3 × 90 ShA and 3 × 50 ShA + 2 × 90 ShA (Figure 7).

The total thickness of the five layers of inserts was constant and amounted to 50 mm. The diameter of the polyurethane inserts was 160 mm. 90MnCrV8 steel dies with the shape shown in Figure 8 were used in the simulations and experiments.

The die was designed to allow for the effect on the sheet formability of differences in the hardness of the polyurethane inserts to be determined. The forming process was intended to test the possibility of creating three independent embosses on one element. The embosses were designed so that the formed material was not significantly pressed to the bottom of the embosses. These embosses did not have technological slopes and were created at an angle of 90° to the die surface. This shape was intended to cause the sheet metal to lift in the areas between the embosses (Figure 8), due to the lack of slopes on the side wall of the embosses. The area of the lower part of the embosses was not finished with a rounding radius, so it was impossible to accurately reproduce the shape of the die. In this way, it was possible to determine the differences between the elements formed with the polyurethane inserts with different hardness configurations.

In the outer area of the formed element, tensile stresses were assumed to occur towards the centre of the die with small compressive stresses caused by the rounded shape of the formed element. This stress varied depending on the area of the element, with the highest value of compressive stress occurring in the die axis, transverse to the length of the emboss. There was a plane tensile stress state in the formed area (Figure 9). However, in the area between the embosses, there was a tensile stress in two axes parallel to the die surface.

### 2.4. Experimental Forming Procedure

To perform tests on the sheet forming of the Inconel 625 alloy sheets, a stamping die was designed based on the Guerin process. The Guerin process was named after Henry Guerin, who, in the late 1930s, discovered the technique of using rubber as the half die instead of the metallic part. This process is commonly used to form short runs of complicated components, such as aircraft panels and automobile panels [28]. Owing to the character of the forming process and the need for the frequent replacement of the polyurethane inserts, the device was developed in accordance with the author’s concept of enabling the quick replacement of sets of components, such as the die, the workpiece and a set of polyurethane inserts. An experimental verification of the simulation results was carried out by performing forming tests using selected combinations of elastomeric materials on a hydraulic press Hydromega 150 T (Hydromega, Gdynia, Poland) with a capacity of 1500 kN. The set of tools that was installed on the hydraulic press is shown in Figure 10.

The top plate with the punch was attached to the upper part of the press. The bottom plate with accessories installed was attached to the worktable of the press. The ejector was pulled out using an actuator located under the worktable. The die, a workpiece and the polyurethane inserts of a given hardness were placed on the base of the pushing system. Special chamfering in the upper part of the guide sleeve positioned all elements relative to each other. Then, the upper cylinder of the press lowered the upper plate together with the punch, exerting force on the polyurethane punch, which caused the workpiece to be deformed. Figure 11 shows a photograph of the tool set mounted on the press.

Owing to the high friction occurring during forming, H-336 lubricant (Molydal, Saint-Maximin, France) was used. It is an oil intended for the plastic forming of sheet metals. It was used to reduce friction between the polyurethane sample and the sheet metal and between the sheet metal and the die surface.

### 2.5. Three-Dimensional Optical Scanning Research Methodology

Three-dimensional optical scanning investigations of the drawpiece geometry were performed using an Atos Core 200 3D scanner (Carl Zeiss GOM Metrology GmbH, Braunschweig, Germany). This device allowed for the creation of three-dimensional models of the analysed elements with a measurement accuracy of 0.017 mm. Thanks to the use of a scanner, 3D models of all the constituent variants were created. A sample scanning result is shown in Figure 12.

The maximum forming depth was determined through a point measurement of the distance between the area with the greatest depression and the reference plane formed on the face of the drawpiece. The measurement was performed to minimise the influence of differences in surface flatness on the measurement of the forming depth. The scans of the drawpiece geometry were used to measure the parameters, such as the surface flatness, thickness of the formed sheet, change in workpiece diameter and the uniformity of the forming.

For the surface flatness measurement, the best plane-matching function was used. This function determined two parallel planes adjusted to the most protruding and the most concave part of the selected surface on the model. The flatness of the surface was determined with the distance between these planes. Figure 13 shows an example result of measuring the surface flatness.

Generating the results of thickness changes (Figure 14a) was possible by determining the normal direction of the scanned surfaces.

This was obtained by analysing the distance between two surfaces of the model at each of the measured points. The distance measurement was used to measure the change in diameter (Figure 14b). Measurements were performed in the direction parallel and perpendicular to the shaped emboss. The result was presented in the numerical form as a reference value determined using the difference between the diameter of the workpiece before the forming process and the measured value.

The uniformity of the forming was assessed by comparing the right side of the model with the left side. The best fit function was used to compare the results, which allowed for the accurate matching of the two models. The result was presented using a colour map of deviations, with point markers placed in key places in the model (Figure 15a).

The difference in the forming depth for the selected polyurethane insert configurations was determined by superimposing 3D scans using the best fit function. To minimise the impact of the emboss depth on the degree of matching, the matching was performed for flat parts of the emboss (Figure 15b). Results for the drawpieces produced under specific conditions were presented in Section 3.

## 3. Results and Discussion

### 3.1. Mechanical Properties of Sheet Metals

Figure 16 shows the tensile curves of the Inconel 625 sheets.

The results included three data series for samples cut at 0°, 45° and 90° angles to the rolling direction of the sheet. A summary of the results of the basic mechanical parameters and anisotropy coefficients is presented in Table 3 and Table 4, respectively.

The results were averaged from three replicates. Those obtained during the tensile test did not show significant differences depending on the orientation of the samples. The highest value of ultimate tensile strength (966.1 MPa) was obtained for a sample oriented parallel to the direction of the sheet rolling. The difference in results depending on the direction of sampling was 11.7 MPa for the tensile strength, 10.9 MPa for the yield stress and 2.6% for the elongation.

### 3.2. Mechanical Properties of Polyurethane Materials

The nominal hardness of the tested materials was verified by performing five measurements for each type of polyurethane used. The results of the individual measurements and the average hardness are presented in Table 5.

In the case of materials with nominal hardnesses of 50, 80 and 90 ShA, the results indicated slight differences depending on the measurement location, not exceeding 2 ShA, which meant that the hardness of the polyurethane materials was uniform. The average value differed significantly for materials with a nominal hardness of 70 ShA and was close to a hardness of 60 ShA.

Figure 17 shows the results of the compression measurements for all the tested polyurethane materials.

These results were used to determine the material constants in the Mooney–Rivlin material model. The compression results showed trends in increasing compressive strength with increasing hardness of the elastomeric material. For samples with a nominal hardness of 60 ShA and 70 ShA, there was a slight difference in the results obtained from the uniaxial compression test.

### 3.3. Wear Resistance

Forming sheets composed of difficult-to-deform alloys using elastomeric materials of much lower hardness results in intensified wear [23,50]. Therefore, determining the quantitative wear of the friction pair is key to establishing the optimal conditions for the forming process. Table 6 shows the wear results of the polyurethane and Inconel 625 nickel alloy samples.

A common feature of the tested polyurethanes was the deformation of their surface in contact with the sample, resulting from the applied force of *F*_N_ = 50 N. The value of this deformation influenced the size of the tribological contact surface in the friction node. Polyurethanes, constituting a countersample, were an element of the friction node, in which mass losses were observed. However, some samples composed of nickel alloy were characterised by an increase in mass resulting from the transfer of the countersample material. For the tested polyurethanes, the COF ranged between 0.263 and 0.622. The polyurethane with a hardness of 90 ShA was characterised by the lowest average COF value. The obtained result of 0.263 was quite a high COF in relation to other polymer materials.

### 3.4. Results of Numerical Simulations of the Forming Process

During the simulation, a forming force of 400 kN was applied to the surface of the polyurethane punch. Figure 18 shows the deformation of the drawpiece in subsequent stages of the simulation.

Table 7 presents the simulation results of all variants of polyurethane insert systems.

The results shown in Table 7 were presented in the form of perpendicular and transverse views, which allowed for an easier assessment of the obtained forming effects. A separate column contains the maximum measured depth of the emboss, which was the main parameter analysed when comparing the tested configurations of polyurethane inserts. The forming depth was chosen as the most important result of numerical computations owing to the easily verified value obtained in numerical modelling, with the results determined during experimental tests. The highest values of sheet metal deformation in the emboss zone (6.124 and 6.4 mm) were obtained for hybrid configurations of the polyurethane inserts with varying hardnesses. It was found that polyurethane inserts of lower hardness in contact with the surface of the sheet being deformed provided greater degrees of sheet metal deformation in the embosses.

Figure 19 compares the maximum depth of emboss for the individual polyurethane hardness variants.

The shape of the die was developed to compare the sheet-metal-forming process using a geometry in which the formed element had three separate areas formed at the same time. The distance between the embosses was selected so that the deformation of the elastomeric material occurred at several independent areas close to each other. This arrangement intended to determine the influence of individual embosses on the overall deformation of the elastomeric material. In tests using five-layer inserts with the same hardness, an increase in the hardness of a polyurethane insert ensured a greater degree of material deformation in the embosses (Figure 19). The results presented in Figure 20 show the influence of the die geometry on the deformation of the elastomeric material.

The deepest embosses were achieved for configurations containing layered polyurethane inserts with a hardness of 2 × 50 ShA + 3 × 90 ShA. The values obtained using these configurations were 6.4 mm for the combination of two layers with a hardness of 50 ShA placed directly above the surface of the workpiece and three layers with a hardness of 90 ShA placed above inserts with a hardness of 50 ShA. For the variant with a changed arrangement of inserts, a value of 6.124 mm was obtained (Table 7). The least favourable result, which was 4.568 mm, was obtained for the polyurethane tool, which comprised five inserts with a hardness of 50 ShA. The difference between the result obtained for the best and worst combination was 1.832 mm. The use of a configuration of 2 × 50 ShA + 3 × 90 ShA allowed for a 40.1% greater stamping depth than the combination containing five inserts with a hardness of 50 ShA.

Numerical simulations were also performed for the forming force, which was increased to 1000 kN. Despite such a high force, no significant differences were observed to depend on the hardness of the polyurethane inserts. Figure 21 presents a summary of sheet metal deformations during the forming with polyurethane inserts with a hardness of between 50 and 90 ShA at a pressure of 1000 kN.

Owing to the lack of clear differences in the simulation results with a forming force of 1000 kN, it was decided to use the simulation results obtained with a forming force of 400 kN to select the sets of elastomer inserts intended for the experimental verification. For inserts of the same hardness, the smallest depth of emboss was obtained for inserts with a hardness of 50 ShA (Figure 19). However, for inserts comprising layers with varying hardnesses, the greatest depth of emboss was provided using inserts with the configuration 2 × 50 ShA + 3 × 90 ShA (Figure 19).

### 3.5. Results of Experimental Tests of the Forming Process

Table 8 presents a summary of the results obtained by forming drawpieces on a hydraulic press with a pressure of 1000 kN depending on the arrangement and hardness of the polyurethane inserts.

This table includes photos of the samples, a 3D scan view and the measured maximum forming depth for each variant. The difference in forming depth for the analysed configuration variants of polyurethane inserts was 0.14 mm. For the variant containing five inserts with a hardness of 50 ShA, the maximum forming depth was 5.81 mm, and for the configuration 2 × 50 ShA and 3 × 90 ShA, it was equal to 5.95 mm (Table 8).

Figure 22 and Figure 23 show the results of the geometric analysis of the drawpieces.

In the case of the results presented in Figure 22, a configuration containing five inserts with a hardness of 50 ShA was used, which, in numerical analyses, showed the worst ability to form an element composed of the Inconel 625 alloy. Figure 23 shows the results for the same forming conditions, but using a configuration containing two inserts with a hardness of 50 ShA and three inserts with a hardness of 90 ShA. In the numerical calculations, this variant showed the best ability to form an element. In the case of the surface flatness, there was a difference of 0.15 mm for both variants considered. For the 5 × 50 ShA variant, the surface flatness was 4.32 mm, and for the 2 × 50 ShA + 3 × 90 ShA configuration, it was 4.17 mm. In the case of the five inserts with a hardness of 50 ShA, there was a uniform thickness distribution.

The difference between the maximum and minimum value was 0.2 mm, which gave a maximum wall thinning of 19%. In the case of the 2 × 50 ShA + 3 × 90 ShA configuration, the thinning was smaller. The minimum wall thickness value was 0.91 mm, which gave a wall thinning of 14%. The change in the diameter of the formed workpiece in the case of the 5 × 50 ShA system was –0.4 mm longitudinally in the direction of the formed embosses and –4.9 mm in the transverse direction. These values for the 2 × 50 ShA + 3 × 90 ShA configuration were –0.69 mm and –4.24 mm, respectively. For the 5 × 50 ShA system, clear differences in the uniformity of formation could be observed. There was a difference of 0.16 mm between the right and left sides of the formed drawpiece in the smaller emboss. For the 2 × 50 ShA + 3 × 90 ShA configuration, a much greater uniformity of formation could be observed, with a maximum difference of 0.06 mm. Figure 24 shows a tabular summary of the measurement results of the drawpieces.

Figure 25 shows a comparison of the 3D scans of measured drawpieces obtained by using two different configurations of polyurethane inserts.

The results confirmed the deeper formation of embosses for the 2 × 50 ShA + 3 × 90 ShA configuration of polymeric inserts. The differences ranged from –0.18 mm to 0.13 mm on the surface of the embosses. Analysing the results of the combinations of inserts, it was possible to conclude that there was a significant difference in the quality of the obtained element depending on the hardness of the elastomeric material used. The results obtained during the numerical simulations were also confirmed experimentally, and for all the measured values of geometric parameters, the 2 × 50 ShA + 3 × 90 ShA configuration was more favourable.

## 4. Conclusions

This article presented the results of numerical and experimental research aimed at determining the influence of the hardness of layered tools composed of polyurethane inserts on the possibility of forming drawpieces from Inconel 625 Ni-based alloy sheets. Based on the research performed, the following conclusions could be formulated:The hardness of the elastomeric materials and configuration of the hardness of layers in a multilayer polyurethane punch had a significant impact on the geometric quality of the obtained drawpieces.An analysis of the results of the numerical simulations and experimental tests clearly indicated that the hardness of the punch material should be selected based on the shape of the formed element.For most of the polyurethane insert configurations, the depth of embosses in the Inconel 625 alloy sheet was similar, at approximately 5.95 mm. The value of the forming depth of emboss was 5.81 mm only for the system containing five inserts with a hardness of 50 ShA.The characteristics of the forming process determined the greatest wear of polyurethane samples on the contact surface with the surface of the formed sheet. The use of multilayer systems of polyurethane inserts allowed for the cost of polyurethane tools to be reduced, owing to the possible replacement of only a single layer, which was in contact with the sheet metal surface, without the need to replace the entire polyurethane tool. It was also possible to rearrange the order of inserts after a certain, precisely defined number of formed drawpieces to ensure uniform tool wear.The determined and verified constitutive models as well as the obtained test results could be used in the design of forming processes in industrial conditions of Inconel 625 nickel alloy sheets using polyurethane inserts of various hardnesses.The results confirmed that when using the designed device, it was possible to form sheets of Inconel-type nickel alloys with a thickness of 1 mm, the forming of which in industrial cold forming conditions is a challenge. Owing to the nature of the forming process, the shaped components could not be heated to high temperatures because of the contact of the polyurethane punch surface with the formed material.

## Figures and Tables

**Figure 1 materials-17-00085-f001:**
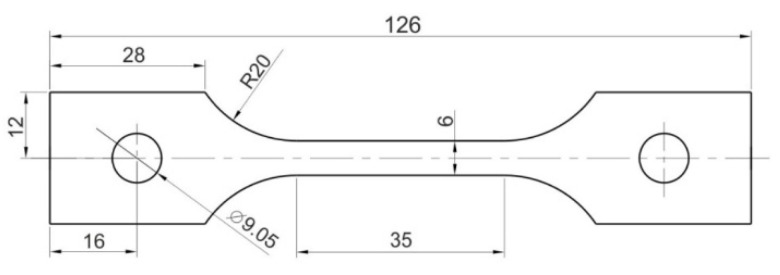
Dimensions of uniaxial tensile test specimens (unit: millimetre).

**Figure 2 materials-17-00085-f002:**
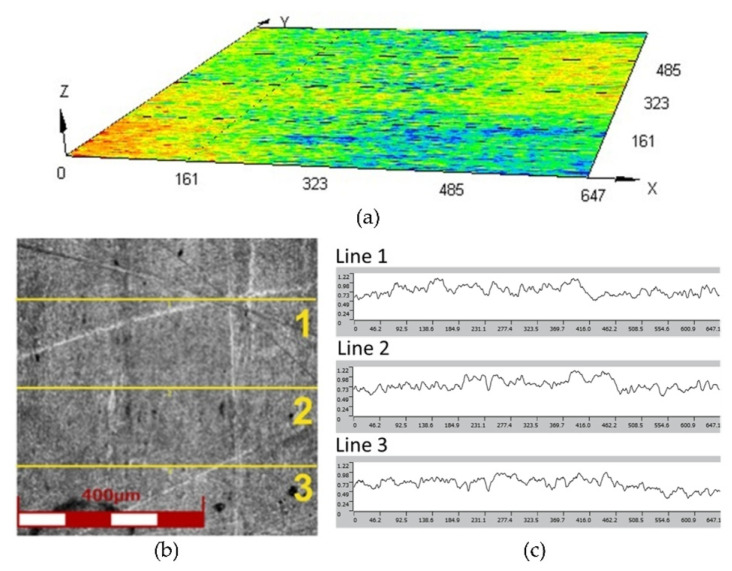
(**a**) Surface topography, (**b**) view of surface of the Inconel 625 sheet metal and (**c**) linear surface roughness profiles used to determine average value of surface roughness parameters.

**Figure 3 materials-17-00085-f003:**
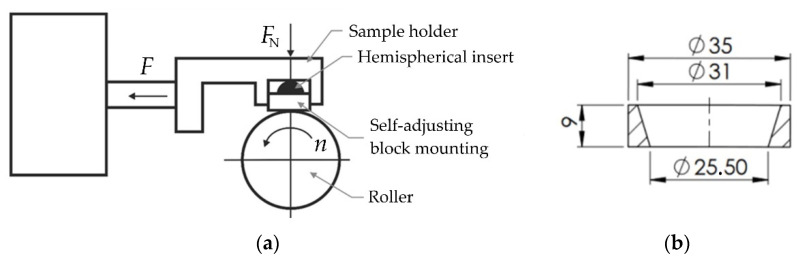
(**a**) Principle of operation of the T-05 tester and (**b**) geometry and dimensions of the polyurethane countersample (unit: millimetre).

**Figure 4 materials-17-00085-f004:**
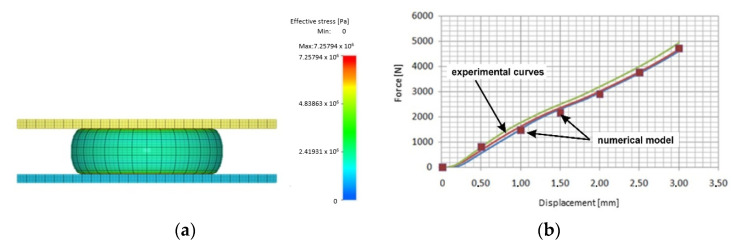
(**a**) Simulated shape of the sample during the compression test and (**b**) fit of the experimental compression curve with the simulation results.

**Figure 5 materials-17-00085-f005:**
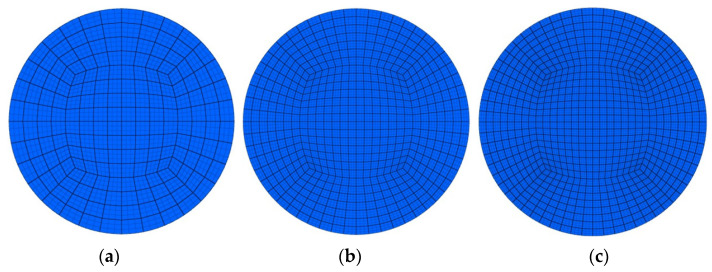
Examples of finite element sizes used in the analysis: (**a**) type of coarse mesh with 8-node hexahedra elements; (**b**) type of medium mesh with quadratic 27-node hexahedron elements; (**c**) type of fine mesh with cubic 64-node hexahedron elements.

**Figure 6 materials-17-00085-f006:**
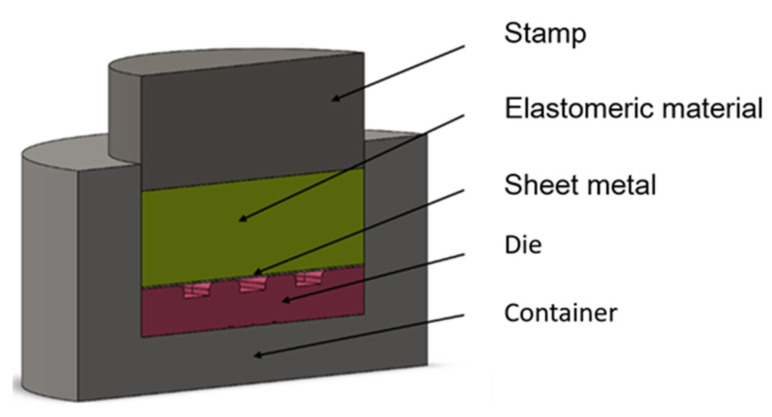
Cross-section of a 3D model of a stamping die used for numerical simulations.

**Figure 7 materials-17-00085-f007:**
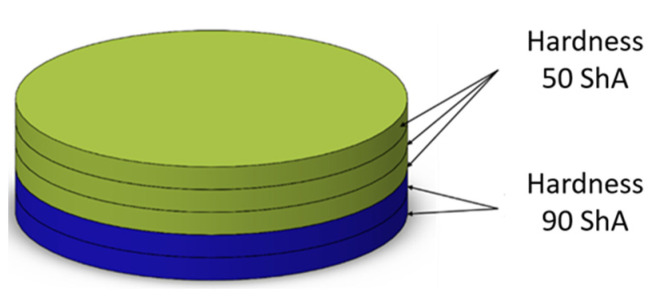
Polyurethane punch configuration with five layers of inserts of different hardnesses (3 × 50 ShA + 2 × 90 ShA).

**Figure 8 materials-17-00085-f008:**
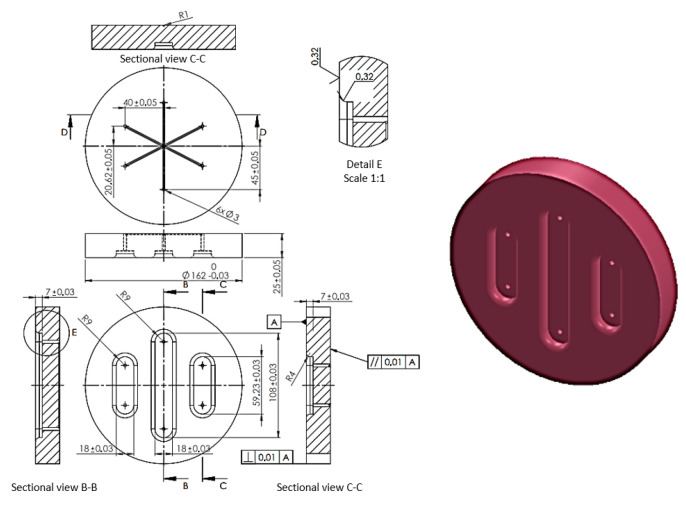
Shape and dimensions of the die used to form Inconel 625 sheet metals (unit: millimetre).

**Figure 9 materials-17-00085-f009:**
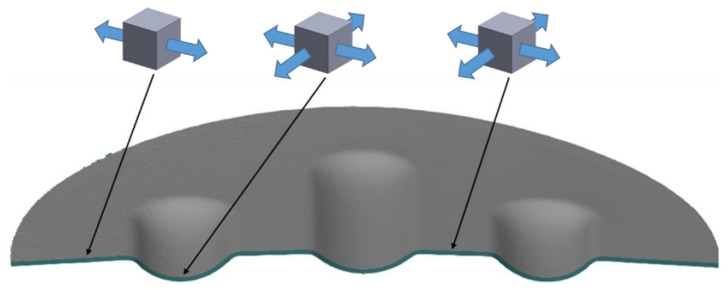
Cross-section of a formed element with marked stress states in selected areas.

**Figure 10 materials-17-00085-f010:**
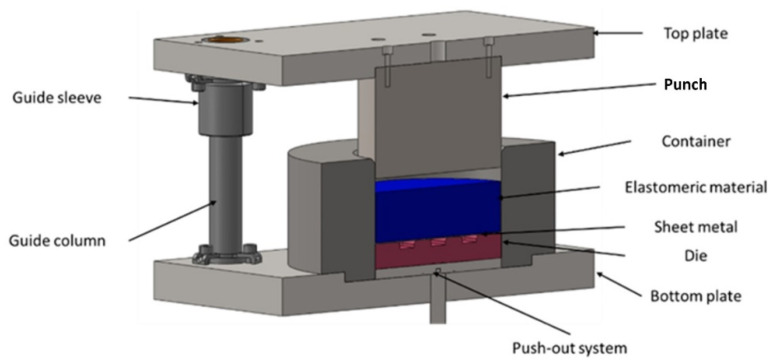
Three-dimensional model of the stamping die.

**Figure 11 materials-17-00085-f011:**
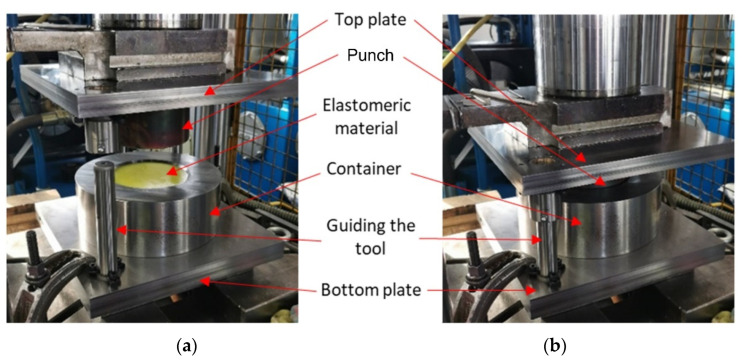
Stamping die (**a**) mounted on a press and (**b**) during forming.

**Figure 12 materials-17-00085-f012:**
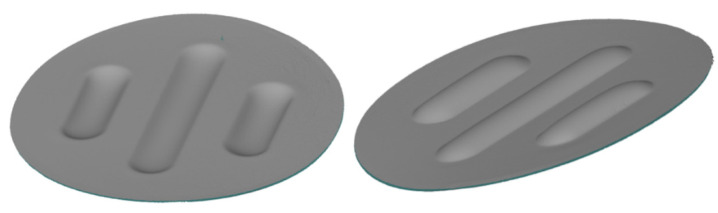
The 3D scan of the drawpiece from different perspectives.

**Figure 13 materials-17-00085-f013:**
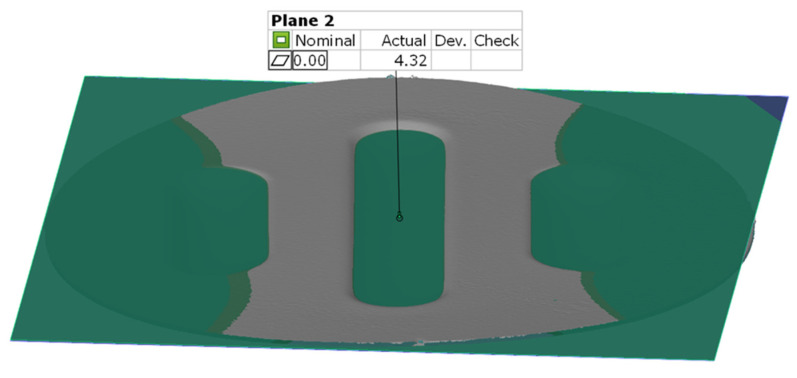
Example result of the surface flatness measurement.

**Figure 14 materials-17-00085-f014:**
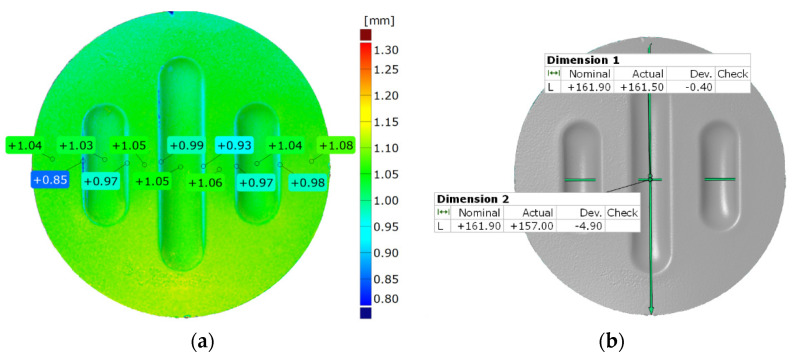
Example result of measuring (**a**) the thickness of the drawpiece and (**b**) the change in drawpiece diameter after the forming process.

**Figure 15 materials-17-00085-f015:**
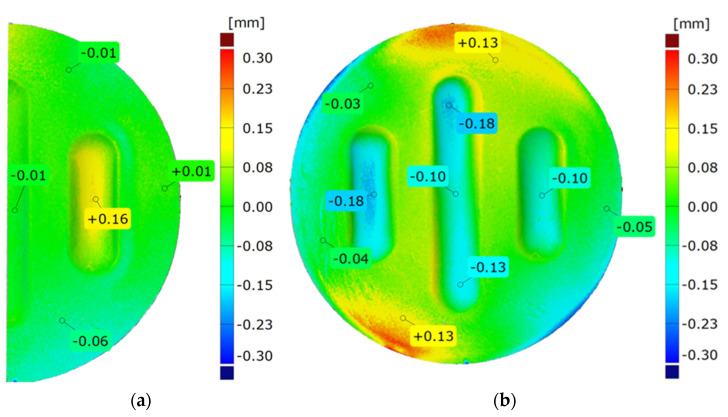
Example results for (**a**) the strain uniformity and (**b**) the difference in forming depth.

**Figure 16 materials-17-00085-f016:**
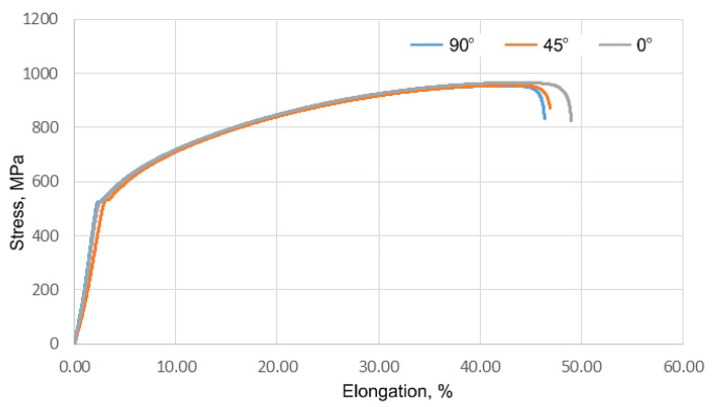
Tensile curves of samples composed of Inconel 625 sheet.

**Figure 17 materials-17-00085-f017:**
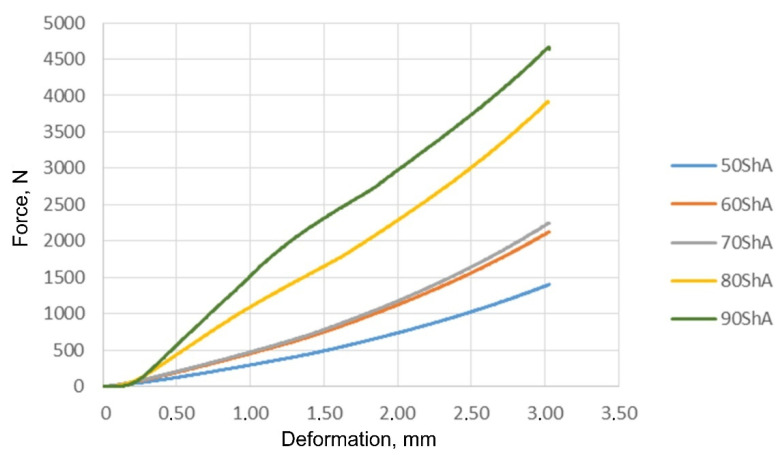
Results of the uniaxial compression test of polyurethane samples.

**Figure 18 materials-17-00085-f018:**
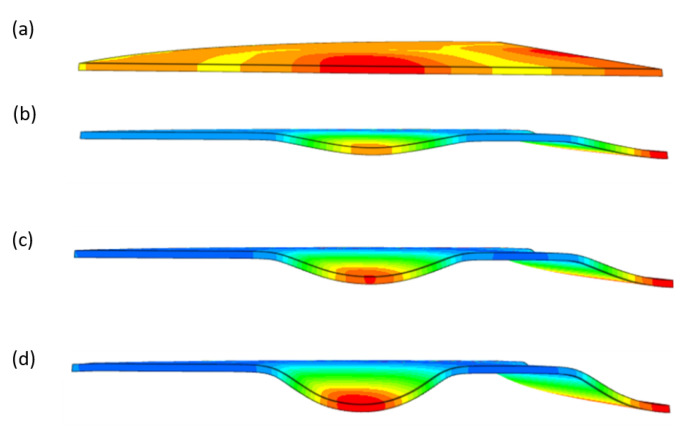
Subsequent stages of forming the element for (**a**) 20%, (**b**) 40%, (**c**) 60% and (**d**) 90% of simulation time.

**Figure 19 materials-17-00085-f019:**
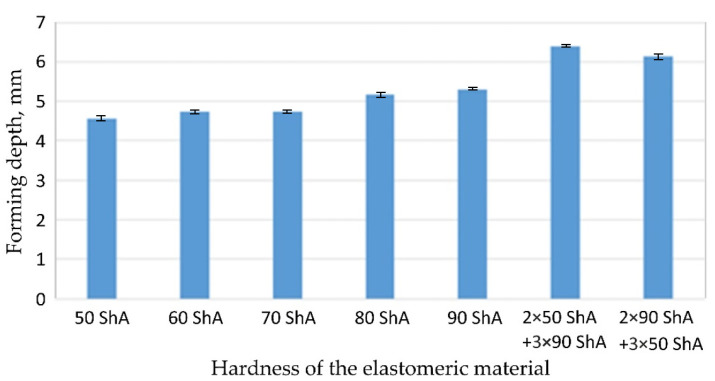
The influence of the hardness of polyurethane inserts on forming depth.

**Figure 20 materials-17-00085-f020:**
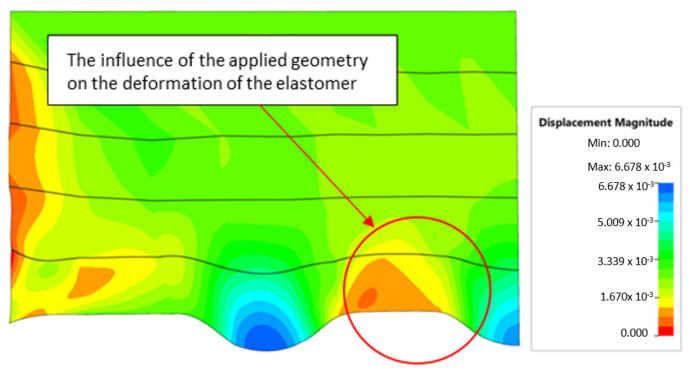
The influence of die geometry on the deformation of the polyurethane punch consisting of five layers of different hardnesses 2 × 50 ShA + 3 × 90 ShA.

**Figure 21 materials-17-00085-f021:**
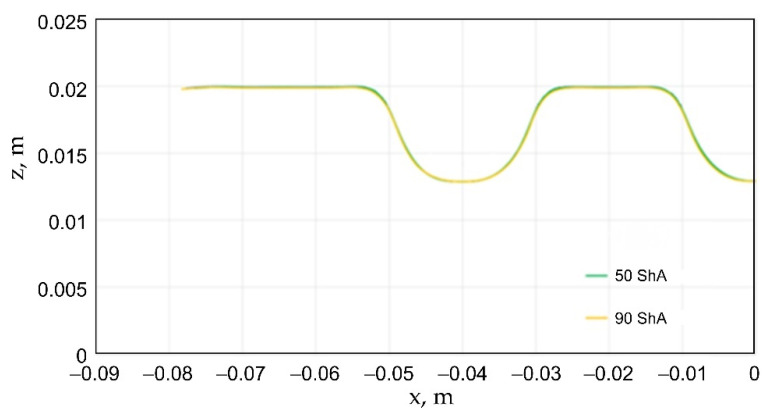
Comparison of the geometry of drawpieces obtained using polyurethane inserts with a hardness of 50 ShA and 90 ShA using a forming force of 1000 kN.

**Figure 22 materials-17-00085-f022:**
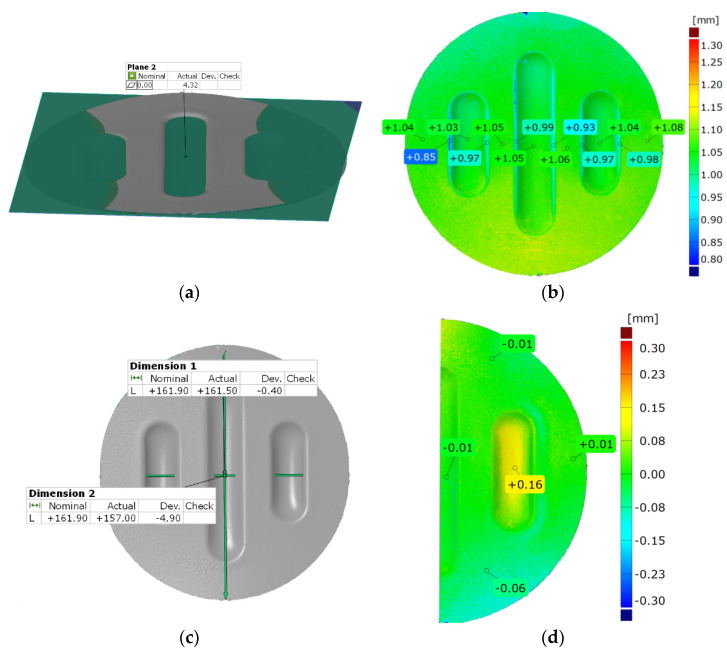
Results of measuring the geometry of the drawpieces formed using five inserts with a hardness of 50 ShA: (**a**) surface flatness, (**b**) thickness, (**c**) changes in the diameter of the workpiece and (**d**) uniformity of forming.

**Figure 23 materials-17-00085-f023:**
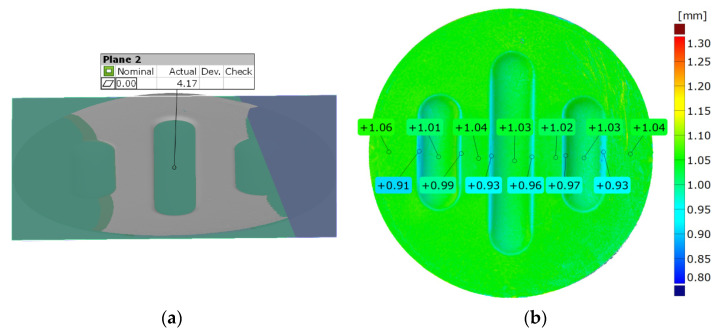
Results of measuring the geometry of the drawpieces formed using two inserts with a hardness of 50 ShA and three inserts with a hardness of 90 ShA: (**a**) surface flatness, (**b**) thickness, (**c**) changes in the diameter of the workpiece and (**d**) uniformity of forming.

**Figure 24 materials-17-00085-f024:**
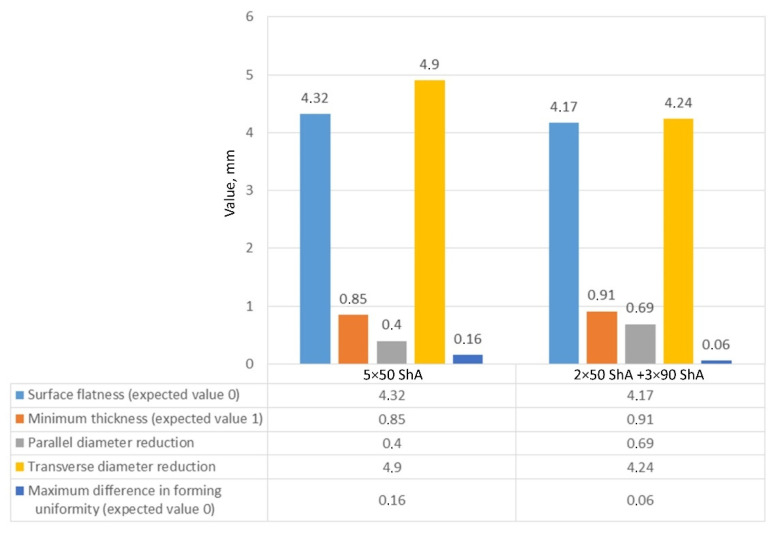
Summary of measurement results of the geometry of drawpieces composed of Inconel 625 sheet metal.

**Figure 25 materials-17-00085-f025:**
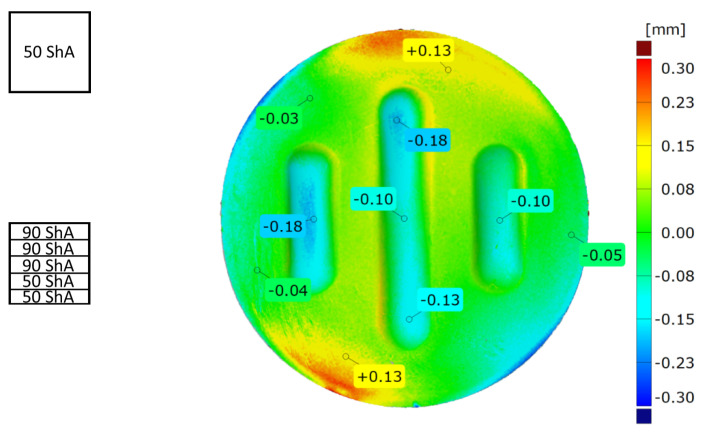
Comparison of the drawpieces formed using two different configurations of polyurethane inserts: 5 × 50 ShA and 2 × 50 ShA + 3 × 90 ShA.

**Table 1 materials-17-00085-t001:** Surface roughness parameters of polyurethane inserts.

Average Value of the Surface Roughness Parameter	Hardness of the Polyurethane Insert				
	50 ShA	60 ShA	70 ShA	80 ShA	90 ShA
Ra, μm	0.337 μm	0.191 μm	0.204 μm	1.228 μm	0.894 μm
Rz, μm	2.536 μm	1.244 μm	1.517 μm	7.918 μm	5.493 μm

**Table 2 materials-17-00085-t002:** Material coefficients for polyurethane samples with varying hardnesses.

Hardness of Polyurethane Sample	Coefficient *K*, Pa	Coefficient *C*_1_, Pa	Coefficient *C*_2_, Pa
50 ShA	4.0 × 10^9^	0.3 × 10^6^	0.15 × 10^9^
60 ShA	4.1 × 10^9^	0.59 × 10^6^	0.19 × 10^6^
70 ShA	4.2 × 10^9^	0.6 × 10^6^	0.2 × 10^6^
80 ShA	4.8 × 10^9^	1.6 × 10^6^	0.11 × 10^6^
90 ShA	4.8 × 10^9^	2.1 × 10^6^	0.1 × 10^6^

**Table 3 materials-17-00085-t003:** Results of the basic mechanical parameters of Inconel 625 sheet.

Sample Orientation	Ultimate Tensile Stress R_m_, MPa	Yield Stress R_p0.2_, MPa	Elongation A, %
0°	966.1 (±5)	526.5 (±3)	49.4 (±2)
45°	955.9 (±4)	532.5 (±4)	47.4 (±2.5)
90°	954.4 (±4)	521.6 (±5)	46.8 (±3)

**Table 4 materials-17-00085-t004:** Anisotropy coefficients of Inconel 625 sheet.

Coefficient of Normal Anisotropy	Coefficient of Planar Anisotropy
1.22 (±0.05)	−0.64 (±0.05)

**Table 5 materials-17-00085-t005:** Results of the hardness measurements of polyurethane samples.

Nominal Hardness, ShA	Number of Measurement					Measured Average Hardness, ShA
	1	2	3	4	5	
50	50	50	51	51	52	50.7
60	62	61	62	62	64	61.8
70	64	66	63	63	66	65.3
80	78	79	78	80	81	79.3
90	89	89	88	90	90	89.3

**Table 6 materials-17-00085-t006:** Wear parameters and COF of polyurethane countersamples and Inconel 625 sample.

Parameter	Hardness of Polyurethane Countersample				
	50 ShA	60 ShA	70 ShA	80 ShA	90 ShA
Weight loss of polyurethane countersample Δ*m*_cs_, %	0.4031	0.1448	0.1956	0.0116	0.0762
Weight loss of Inconel 625 sample Δ*m*_s_, %	0.1958	−0.041	0.2483	−0.063	−0.0328
Average COF	0.465	0.526	0.622	0.545	0.263

**Table 7 materials-17-00085-t007:** Results of the maximum depth of emboss in the Inconel 625 alloy sheet for individual variants of the polyurethane inserts.

Hardness of Polyurethane Inserts	Results of Displacement	Maximum Depth of Emboss
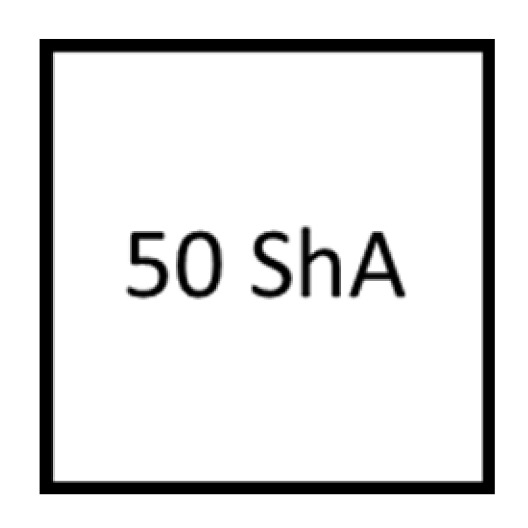	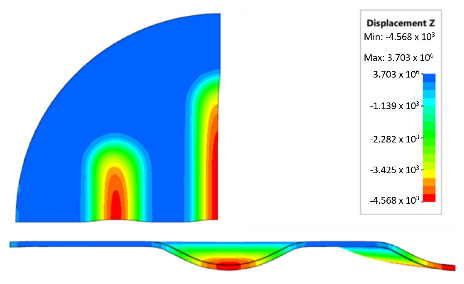	4.568 mm
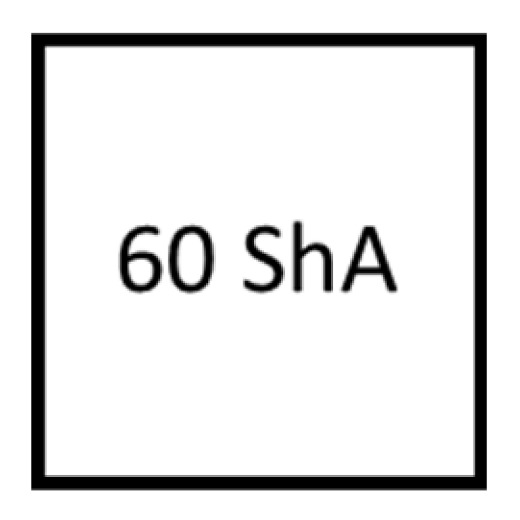	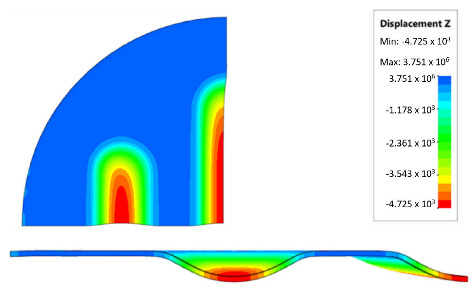	4.725 mm
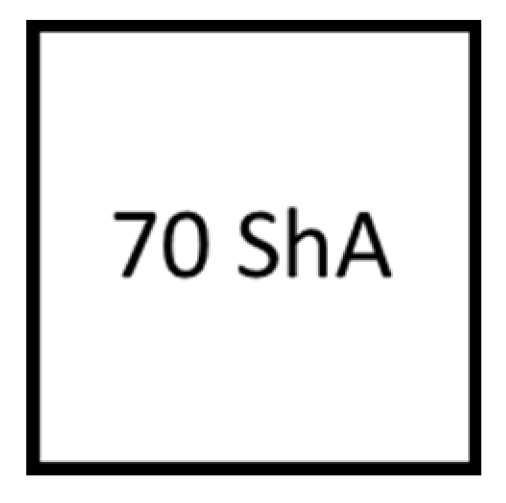	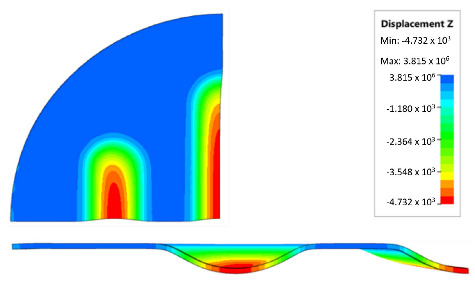	4.732 mm
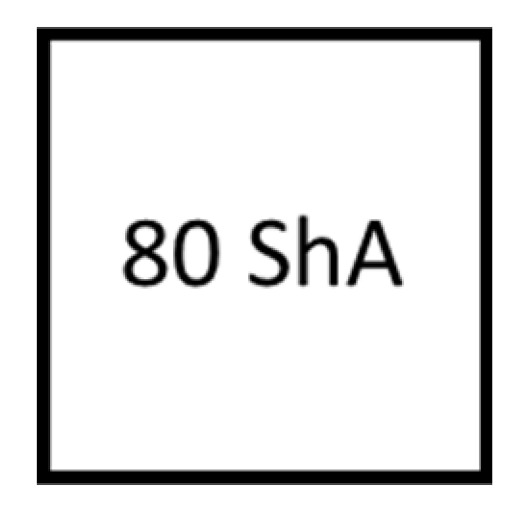	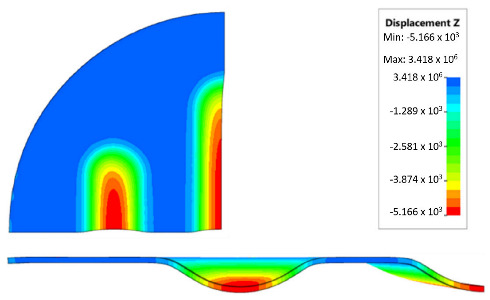	5.166 mm
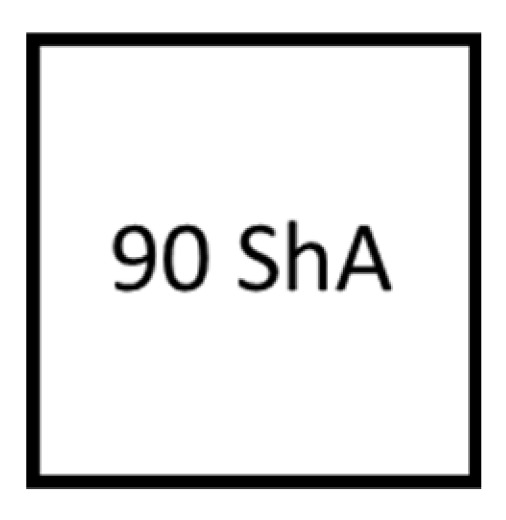	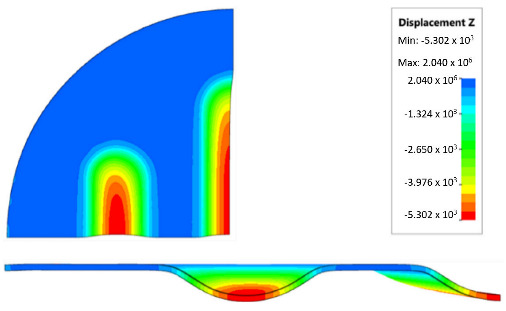	5.302 mm
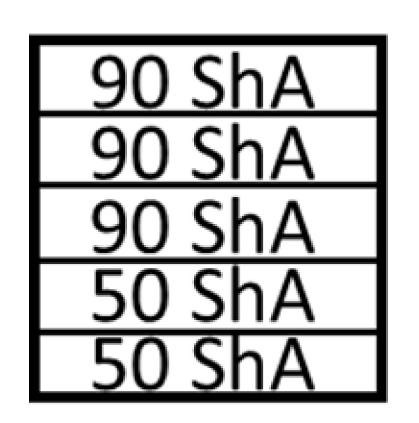	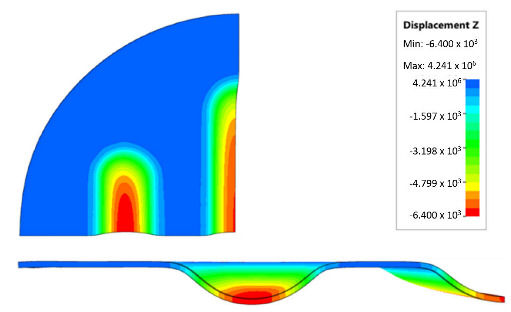	6.4 mm
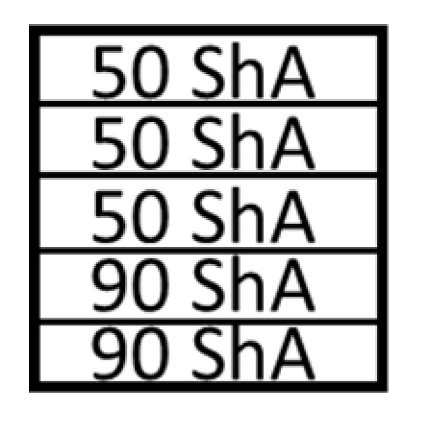	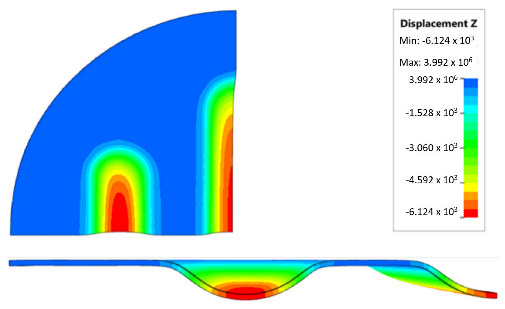	6.124 mm

**Table 8 materials-17-00085-t008:** Summary of the results of forming drawpieces with a pressure of 1000 kN.

Measurement Setup	Image of the Formed Sheet	3D Scan of the Formed Sheet	Maximum Forming Depth
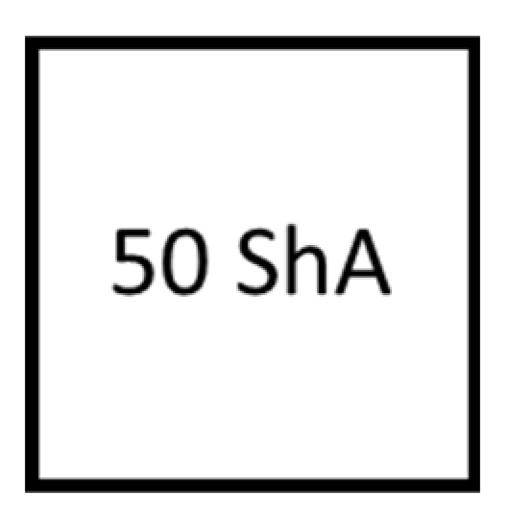	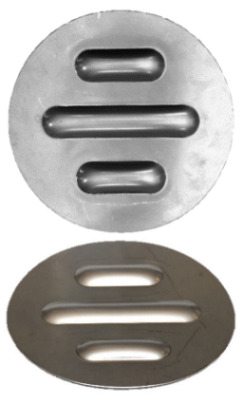	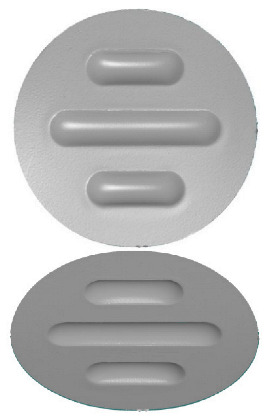	5.81 mm
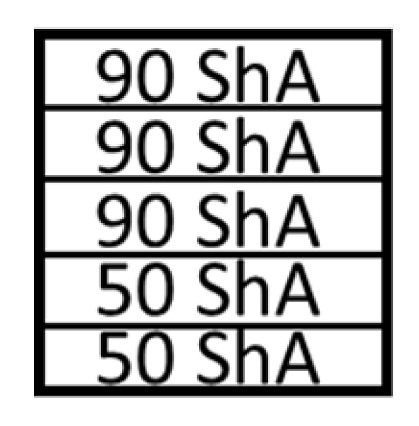	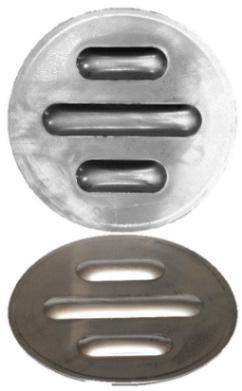	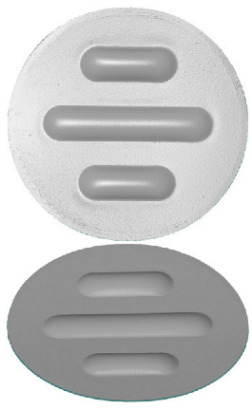	5.95 mm

## Data Availability

Data are contained within the article.

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
