# Peer review of "Experimental Research and Numerical Modelling of the Cold Forming Process of the Inconel 625 Alloy Sheets Using Flexible Punch"

_materials, 2023, doi:10.3390/ma17010085_

Round 1

Reviewer 1 Report

Comments and Suggestions for Authors

In this manuscript, the authors presented a study regarding the use of flexible tools made of elastomeric materials applied to the cold forming process of difficult-to-deform sheet alloys, such Inconel 625. The structure and relevance of the study are clearly identified and the topic, as well as the content, is also within the scope of the journal. However, the following points should be addressed:

1. Some text along the manuscript appear to be in red color. Authors should revise some typos:

Line 268 – “2.5.3. D” should be “2.5. 3D”?

Line 375 - Please replace “The results presented” by “The results shown”.

2. Regarding the uniaxial tensile tests of the Inconel 625 (figure 15), what is the reason for the existence of a non-linear behavior at the beginning of the engineering stress-strain curve?

3. Authors may consider, for all present experimental results, a corresponding uncertainty measurement value, so the readers can have a reference basis of dispersion.

4. Since a mesh sensitivity analysis was made (stated in section 3.5), authors should include a representation of the considered finite elements deformable parts discretization, so that they can be used to complement the analysis of simulation contours results.

5. The numerical simulation results using a forming force of 1 kN do not show clear differences, so for the experimental validation it was consider a forming force of 400 kN. Although, in the experimental test’s validation section (section 3.5), it was used a hydraulic pressure of 1000 kN. Authors should clarify this subject.

Reviewer 2 Report

Comments and Suggestions for Authors

By investigating the influence of the hardness of elastomeric materials on the formation of nickel-based alloys such as Inconel 625, the authors provide valuable information to understand how different material conditions affect the part forming process.

This research could have significant practical implications in the manufacturing industry, especially for those dealing with the forming of difficult-to-deform alloys such as Inconel. The findings can guide the selection of materials and manufacturing methods to improve the quality of the parts produced.

By performing both numerical simulations and experimental studies, the paper contributes to validating theoretical models used in industry to predict the behavior of materials and forming processes. This is essential to ensure the accuracy of simulations used in the design of industrial tools and processes.

The observation of non-uniform deformation of the metal can be crucial in warning about potential challenges in the manufacture of parts made from nickel alloys such as Inconel. Furthermore, conclusions about the influence of elastomer hardness on the geometric quality of parts can provide guidelines for optimizing the process.

By concluding that the selection of polyurethane hardness should be determined by the shape of the formed element, this paper opens the door for future research focused on optimizing material and process parameters for different part shapes.

Reviewer 3 Report

Comments and Suggestions for Authors

This article uses a combination of simulation and experiment to introduce in detail the effects of different polyurethane blade combinations on the forming of Inconel 625 nickel-based alloy sheets. There are several main suggestions:

1. What is the basis for this value of μ = 0.09.

2. When using different combinations of hardnesses of polyurethane inserts for forming, will the friction between the inserts affect the maximum forming depth and wall thickness uniformity of the workpiece under a certain forming force.

3. The reasons of forming law under different insert conditions should be analyzed.
